Temperature-driven dynamics: unraveling the impact of climate change on cryptic species interactions within the Litoditis marina complex

Vafeiadou Anna-Maria annamaria.vafeiadou@ugent.be 1
Geldhof Kevin 1
Barhdadi Wissam 2
Baetens Jan M. 2
De Baets Bernard 2
Moens Tom 1
Daly Aisling J. 2
1 Department of Biology, Marine Biology Research Group, Universiteit Gent , Ghent , Belgium
2 Department of Data Analysis and Mathematical Modelling, Universiteit Gent , Ghent , Belgium
Zhukova Natalia
Electronic publication date: 2024 May 20
Publication date: 2024
Volume: 12
Electronic Location ID: e17324
Received 2024 Feb 9; Accepted 2024 Apr 10
Copyright: ©2024 Vafeiadou et al.
Copyright year: 2024
Copyright holder: Vafeiadou et al.
License: This is an open access article distributed under the terms of the Creative Commons Attribution License, which permits unrestricted use, distribution, reproduction and adaptation in any medium and for any purpose provided that it is properly attributed. For attribution, the original author(s), title, publication source (PeerJ) and either DOI or URL of the article must be cited.
License URL: https://creativecommons.org/licenses/by/4.0/

Keywords: Temperature, Competition, Facilitation, Species interactions, Coexistence, Cryptic species, Climate change, Modeling, Population dynamics

Funding: Ghent University’s Special Research Fund (BOF) 01G02617 BOF-GOA project ‘Assessing the biological capacity of ecosystem resilience’ 2017000601 EMBRC Belgium - FWO international research infrastructure I001621N This work was supported by the Ghent University’s Special Research Fund (BOF) through GOA project no. 01G02617. Wissam Barhdadi was supported by the BOF-GOA project ‘Assessing the biological capacity of ecosystem resilience’ (no. 2017000601). The research was carried out with infrastructure funded by EMBRC Belgium - FWO international research infrastructure (no. I001621N). The funders had no role in study design, data collection and analysis, decision to publish, or preparation of the manuscript.

==============================
Anthropogenic climate change and the associated increase in sea temperatures are projected to greatly impact marine ecosystems. Temperature variation can influence the interactions between species, leading to cascading effects on the abundance, diversity and composition of communities. Such changes in community structure can have consequences on ecosystem stability, processes and the services it provides. Therefore, it is important to better understand the role of species interactions in the development of communities and how they are influenced by environmental factors like temperature. The coexistence of closely related cryptic species, with significant biological and ecological differences, makes this even more complex. This study investigated the effect of temperature on species growth and both intra- and interspecific interactions of three species within the free-living nematode Litoditis marina complex. To achieve this, closed microcosm experiments were conducted on the L. marina species Pm I, Pm III and Pm IV in monoculture and combined cultures at two temperature treatments of 15 °C and 20 °C. A population model was constructed to elucidate and quantify the effects of intra- and interspecific interactions on nematode populations. The relative competitive abilities of the investigated species were quantified using the Modern Coexistence Theory (MCT) framework. Temperature had strong and disparate effects on the population growth of the distinct L. marina species. This indicates temperature could play an important role in the distribution of these cryptic species. Both competitive and facilitative interactions were observed in the experiments. Temperature affected both the type and the strength of the species interactions, suggesting a change in temperature could impact the coexistence of these closely related species, alter community dynamics and consequently affect ecosystem processes and services.

Introduction

Marine environments encompass a significant part of Earth’s biodiversity, providing essential services such as nutrient cycling and food production. Human-driven climate change, evidenced by rising global mean sea surface temperatures and ocean chemistry changes (IPCC et al., 2019), is altering marine ecosystems. These shifts affect benthic community dynamics and species interactions, particularly competition, with potential cascading effects on ecosystem structure and functioning (Kordas, Harley & O’Connor, 2011; Schratzberger & Ingels, 2018; Akesson et al., 2021).

Biodiversity enhances ecosystem processes, supporting productivity and resource use through functional complementarity across various ecosystems (Danovaro et al., 2008; Tilman, Isbell & Cowles, 2014). Diverse communities exhibit enhanced stability due to varied responses to environmental changes and stronger resistance to invaders (Tilman, Isbell & Cowles, 2014). Alongside diversity, community composition is another important determinant of functional diversity. Even small changes in the assemblage of a community can affect the functional trait distribution and can have far-reaching implications for the functioning of the ecosystem and its services (Vafeiadou et al., 2018). Competitive interactions are species-specific and influenced by factors like food availability and dispersal, leading to co-occurrence or exclusion of species (Chesson, 2013; Hart, Freckleton & Levine, 2018; Moens & Beninger, 2018). Classical competition theories propose that closely related species with similar niches face intense competition. This principle of competitive exclusion (Hardin, 1960) suggests that very similar species cannot coexist due to competition, the intensity of which is often associated with species phylogenetic relatedness (Violle et al., 2011). Empirical evidence challenges this principle, however, showing coexistence among functionally similar species (Godoy, Kraft & Levine, 2014; Kaplan & Denno, 2007; Venail et al., 2014; Derycke et al., 2016).

In addition to competition, less-studied positive interactions, commonly classified under facilitation, profoundly impact community structure and diversity (Bruno, Stachowicz & Bertness, 2003; Hay et al., 2004; Mcintire & Fajardo, 2014). Facilitative interactions enable coexistence and are mediated by various mechanisms like habitat creation, stress reduction, resource enhancement, and indirect influences on competitors or predators (Bruno, Stachowicz & Bertness, 2003; Mcintire & Fajardo, 2014). Similar to competition, facilitation is highly context-dependent, driven by biotic and abiotic factors affecting its strength and significance (Bruno, Stachowicz & Bertness, 2003; Bruno & Bertness, 2001).

The Modern Coexistence Theory (MCT) framework offers insights into predicting competitive outcomes, utilizing metrics like the average fitness ratio and niche overlap (Chesson, 2000; Godwin, Chang & Cardinale, 2020). The Lotka–Volterra competition model provides a foundation for understanding competition dynamics (Chesson, 2013; Godwin, Chang & Cardinale, 2020). However, MCT’s applicability is limited to pairwise interactions, and incorporating non-competitive interactions remains a challenge (Barabás, D’Andrea & Stump, 2018; Godwin, Chang & Cardinale, 2020).

Free-living nematodes, the most abundant invertebrates in marine sediments and in sediments and soils in general, are known to play an essential role in the decomposition of organic matter, nutrient regeneration, and energy transfer to higher trophic levels (Coull, 1999; Moens et al., 2013; Schratzberger & Ingels, 2018). Due to their widespread distribution and high abundance, they serve as excellent study organisms to explore population, community, and biodiversity dynamics. Their relatively short generation times, high reproduction rates, and adaptability to lab conditions make them valuable models for studying environmental effects (Schratzberger & Ingels, 2018). In addition, behind the morphological diversity of many free-living nematodes hides a substantial cryptic diversity. This is, for instance, the case for our model (cryptic) species (complex) Litoditis marina (Bastian, 1865) Sudhaus, 2011 (formerly Rhabditis marina or Pellioditis marina). L. marina comprises at least 10 cryptic species (Pm I–Pm X), which exhibit some variability in morphology, reproductive behavior, and physiological tolerance across their range (Derycke et al., 2005; Fonseca, Derycke & Moens, 2008; Moens & Vincx, 2000a). Some occur sympatrically with up to four cryptic species in the same (micro)habitats (Derycke et al., 2006). All L. marina cryptic species are effective colonizers of organically enriched coastal habitats due to their rapid development and high reproductive potential when food is plentiful (Derycke et al., 2006; Moens & Vincx, 2000a).

The co-occurrence of two or more cryptic species of the L. marina species complex on a small spatial scale appears at odds with the traditional competitive exclusion principle (Violle et al., 2011). Their coexistence primarily arises from niche partitioning, driven by subtle differences in resource use or environmental preferences (De Meester et al., 2011; De Meester et al., 2015a; Derycke et al., 2016; Guden et al., 2018; Guden, Derycke & Moens, 2021). Neutral dynamics (Hubbell, 2005) or fluctuating environmental conditions also contribute to coexistence, as do temporal fluctuations in biotic and environmental factors (Daly et al., 2021; De Meester et al., 2011; De Meester et al., 2015b). Occasional episodic occurrence of specific environmental conditions can temporarily favor competitively inferior species, and facilitate coexistence with stronger competitors (Guden et al., 2018). Complex non-transitive competition networks where competitive hierarchies depend on the exact identity of the component species, may further contribute to multispecies coexistence (De Meester et al., 2015c).

Changes in environmental conditions like temperature can influence the interactions between species, leading to cascading effects on the abundance and composition of communities. The MCT framework can be used to estimate the relative competitive abilities of different species under changing conditions. Understanding how species interactions and their role in community development are influenced by increasing temperature, can provide insights into future changes to marine communities and ecosystems. Here we focused on three species within the cryptic L. marina complex: Pm I, Pm III and Pm IV, to address the following research questions: (i) Does temperature differentially affect population growth in co-occurring L. marina species? (ii) Does temperature affect the interspecific interactions between L. marina species? 

Materials & Methods

Cultures

Experiments were performed on three cryptic species of the L. marina complex: Pm I, Pm III and Pm IV. Nematode specimens were harvested from monospecific cultures in the exponential growth phase of the populations grown on agar medium. Monospecific cultures were initiated from single gravid females collected and isolated from macroalgal substrates, i.e., Fucus sp. stands from the Paulina mudflat at the Western Scheldt estuary, the Netherlands (Pm I and Pm III), and Ulva sp. stands from Lake Grevelingen, the Netherlands (Pm IV). They were identified at the species level using a dedicated qPCR protocol (Derycke et al., 2012). Stock cultures were maintained on sloppy agar medium (1%, 4/1 ratio of bacto to nutrient agar) prepared with sterile artificial seawater (ASW) at a salinity of 25 (Moens & Vincx, 1998). The pH was maintained at 7.5–8 by adding TRIS-HCl buffer at a final concentration of 5 mM. The stock cultures were maintained in the dark at 18 °C, a temperature comparable to the average summer temperature of the Westerschelde estuary (Depreiter et al., 2014). Cultures were refreshed weekly and supplied with 100 µL of frozen-and-thawed Escherichia coli (strain K12) suspension of 3 × 109 cells mL−1. A small amount of cholesterol (10 mg L−1) was also added as a source of sterols since nematodes on a strictly bacterial diet are unable to synthesise sterols de novo (Vanfleteren, 1980).

Experimental setup

The experiments comprised three monospecific treatments (referenced as treatment codes M1, M3 and M4) to establish the population dynamics in the absence of competition for the species Pm I, Pm III and Pm IV, respectively, and three combined treatments to investigate the pairwise interactions between Pm I, Pm III and Pm IV (referenced as C13, C14 and C34). To investigate the effects of temperature on species interactions, each of these monospecific and combined-species treatments was incubated at two constant temperatures: 15 °C and 20 °C. These represent average daytime temperatures during early autumn and late spring (15 °C) and summer (20 °C) at the locations of origin of the nematode cultures (Depreiter et al., 2014).

Four replicates per combination of interaction treatment and temperature were prepared in round Petri dishes (8.4 cm int.diam.), filled with 12 mL of a 1% bacto-nutrient agar (4:1 w:w) (see Cultures section). These plates were supplied with 100 µL of an E. coli suspension (3 × 109 cells mL−1) at the start of the experiment. After 14 days of incubation, all populations were transferred to larger Petri dishes (14 cm int. diam.) filled with 29 mL agar medium and provided with 242 µL of the E. coli suspension. The amount of food offered was proportional to the volume of agar medium in the experimental microcosms, which in turn was roughly proportional to a microcosm’s surface area. Each plate of the monospecific treatments was inoculated with three adult males and five adult females. Healthy and vivid individuals were handpicked from the stock cultures and randomly placed in the Petri dishes. The selected females were all at the start or in an early phase of their reproductive period. Four replicates were prepared per species and temperature, resulting in a total of 24 monospecific cultures. Every four days (which corresponds to one full development time of L. marina at a temperature of 20 °C), adult nematodes of all replicates were counted under a stereomicroscope. This resulted in a total of 6 time points during the incubation period of 24 days.

The interaction (combined species) experiment was performed using an additive design, so that every replicate of the multispecies treatments was inoculated with three adult males and five adult females of each species. To determine the species composition of the multispecies assemblages after a certain amount of time, the plates were frozen at −20 °C for further species identification using qPCR. Because of the destructive sampling for qPCR, 4 separate replicates were prepared for every time point (of 6 time points), which resulted in a total of 144 plates for all treatments. The total numbers of adult nematodes in the competition treatments were counted at each time point using a stereomicroscope, after which plates were kept frozen at −20 °C prior to qPCR identification. The absolute abundance of each species was then calculated by multiplying the total absolute abundance of the populations with the relative abundance determined by real-time qPCR (see below).

Quantification of relative abundance

The relative abundance of each cryptic species in the competition assays was determined using qPCR (Derycke et al., 2012; De Meester et al., 2015b). The agar was melted by transferring it into 1 L of distilled water at 70 °C, allowing the extraction of all nematodes by a simple sieving protocol over two sieves with respective mesh sizes of 125 µm and 32 µm. By doing so, adults and juveniles were separated based on their body sizes (De Meester et al., 2015b). The adult fraction was then subsampled to obtain approximately 100 nematodes per subsample. Nematode DNA was extracted using the hexadecyltrimethylammonium bromide (CTAB) protocol as modified for L. marina (Derycke et al., 2012).

Relative quantification of the cryptic species was performed using qPCR of the ribosomal ITS region using a Lightcycler 480 System and the LightCycler 480 SYBR Green I master kit (Roche Diagnostics, Basel, Switzerland). Prior to qPCR, DNA was quantified using a NanoDrop 2000 (Isogen Life Science). Samples with DNA concentrations >10 ng µL−1 were diluted with distilled water. The qPCR mixtures were prepared for 10-µL reaction volumes, using 5 µL of the master mix, 3 µL of the species-specific primers (final concentration of 1 µM for Pm I and Pm III, 200 nM for Pm IV), 1.5 µL of PCR-grade water and 0.5 µL of the nematode DNA sample. All samples were run with two technical replicates. A positive control using 0.5 µL of DNA template per species and a negative control (no DNA template) were included. The thermal cycling protocol was initiated by a denaturation step at 95 °C for 10 min and followed by 40 cycles of denaturation (10 s at 95 °C), annealing (20 s at 60 °C) and extension (20 s at 72 °C).

The cycle threshold (CT) values from the qPCR were then used to calculate the relative abundances of each cryptic species using the adjusted Δ ΔCT method (Livak & Schmittgen, 2001; Mommer et al., 2008). Finally, the relative contribution of a species (in units of percent) was calculated by comparing the estimated ratio of the species (ERSpecies) to the sum of the estimated ratio of all species present (ERSum): (1) Estimated species contribution=ERSpeciesERsum∗100.

The estimated species contribution was then multiplied by the total (adult) nematode abundance to derive the absolute abundance of each species.

Competition model

To analyse the effects of temperature and competition on population and assemblage dynamics, the experimental data was used in combination with a simple population model. The set of differential equations was solved numerically using the Runge–Kutta method, written in Python (version 3.8.5). The model was based on the Lotka–Volterra equations for competition between two species. (2a) dN1dt=r1N11−∝11N1−∝12N2

(2b) dN2dt=r2N21−∝22N2−∝21N1.

The variables and parameters involved in Eqs. (2a) and (2b) are described in Table 1.

Table 1 Model variables, parameters and units of the competition model.

Variable	Description	Unit	
N 1	Abundance of species 1	Individuals	
N 2	Abundance of species 2	Individuals	
Parameter			
r 1	Maximum per-capita growth rate of species 1	s−1	
r 2	Maximum per-capita growth rate of species 2	s−1	
α 11	Competition coefficient of intra-specific competition (density dependence) for species 1	Individuals−1	
α 12	Competition coefficient of interspecific competition of species 2 on species 1	Individuals−1	
α 22	Competition coefficient of intraspecific competition (density dependence) for species 2	Individuals−1	
α 21	Competition coefficient of interspecific competition of species 1 on species 2	Individuals−1	

Initially, the parameters involved in intraspecific competition (the maximum per capita growth rate and intraspecific competition coefficient) were determined separately for each species (Pm I, Pm III and Pm IV) by temperature combination. To achieve this, the population data (total abundance over time) from the monospecific experiments of each species and replicate was fitted to an abridged version of Eqs. (2a) and (2b): (3) dNdt=rN1−∝N.

The parameter estimation was performed using a covariance matrix adaptation evolution strategy (CMA-ES) using the cma-package for Python, with the sum of squared residuals (SSR) as the optimised parameter.

The estimated maximum growth rate and intraspecific competition coefficient for each species and temperature were then used to estimate the interspecific competition coefficients for each species pair and temperature combination. This parameter estimation was performed by fitting the average population data from competition treatments with the pairwise Lotka–Volterra model as described above. This means the input data for the monospecific model and multispecies model were different. The input for the monospecific model was the number of adult individuals for each replicate of each species over time. This method was used in order to perform statistical analysis on the estimated parameters. This was not possible in the multispecies model due to a lack of sufficient data for every time point in the experiment (discussed further in Results). Therefore, the input for the multispecies model was the average number of adult individuals for each species over time. The same estimation strategy was used for the multispecies model as for the monospecific model.

Model validation was performed by calculating the root-mean-square error (RMSE) and normalising by the mean (NRMSE). Values closer to zero therefore show a better model fit.

Average fitness ratio

The estimated parameters from both monospecific and multispecies models were used to calculate the average fitness ratio for each species pair. The original formula for average fitness ratio by Chesson (2000) does not account for facilitative interactions, represented mathematically by negative competition coefficients. Since the estimated competition coefficients contain multiple negative values, an adjusted version of the average fitness ratio was calculated using the method of Bimler et al. (2018): (4) κjκi=e∝ije∝iie∝jje∝ji.

This adjusted average fitness ratio will result in a value between zero and + ∞. A value larger than one indicates that species j is the better competitor. A value lower than one suggests species i is the superior competitor in the species pair.

Significance testing

For the monospecific experiments, the effects of temperature on the total abundance of adult populations after 24 days were investigated using a two-way analysis of variance (ANOVA), with temperature and species as main factors, after the assumptions of normality and homoscedasticity were verified. Significant factors were further investigated using Tukey Honest Significant Differences (HSD) tests. Two-way ANOVAs were also performed to investigate the effects of Temperature and species interactions (Combination) on the total abundance of adult populations after 24 days in the combination treatments after assumptions were met. The same tests but on the maximal abundance of adults were also performed per species, for both the monospecific and combination experiments.

Results

Monospecific experiment

Population abundance in the monospecific treatments

Total abundance of adult nematodes in the monospecific treatments after 24 days varied significantly between species, temperatures and their interaction (two-way ANOVA: Temp: F = 27.4; p = 5.6E−05, F = 39.1; Sp: p = 2.79E−07, Temp × Sp: F = 59.2; p = 1.21E−08; Fig. 1).

Figure 1 Maximal abundances of adult individuals per species in monospecific treatments at two temperature regimes: 15 °C and 20 °C.

The number of days until the maximum was attained is shown above each bar. M1, M3, M4 refer to monospecific populations of species PM I, PM III and PM IV, respectively. Data are mean abundances of adult individuals of 4 replicates per treatment and error bars represent 95% confidence intervals.

Maximal or close to maximal abundances were nearly always obtained after the same time interval of 24 days (Figs. 1 and 2), with the exceptions of Pm I and Pm IV which reached their highest abundances after 8 days at 15 °C for Pm IV and after 8 and 16 days at 20 °C, for Pm I and Pm IV respectively. Moreover, Pm I reached higher adult abundances at 15 °C, while Pm III and Pm IV showed their maximal population abundances at 20 °C.

Figure 2 Abundances of adult individuals in monospecific treatments over time for Pm I (top), Pm III (middle) and Pm IV (bottom) at two temperature regimes: 15 °C (left) and 20 °C (right).

Abundances are represented by recorded data (data points) and the fitted model (line). Error bars represent 95% confidence intervals. Note very different scales of the Y-axis in different figure panels.

The logistic population growth model (Eqs. (2a) and (2b)) showed a good fit for most treatments (Fig. 2). Pm I at 20 °C had the worst fit (NRMSE = 0.9739) as the population reached its carrying capacity after 8 days at this temperature, then dramatically declined and ultimately collapsed after 24 days; hence, a logistic growth pattern could not be assumed. The population of Pm IV at 15 °C collapsed after 16 days, therefore, the last two time points were not included for the fitting of the population model for this treatment.

Population growth model in the monospecific treatments

Temperature had a significant effect on the maximum growth rate (r) for all species (two-way ANOVA: Temp: F = 11.02; p = 0.004, Sp: F = 1.03; p = 0.376, Temp × Sp: F = 1.22; p = 0.319), the maximum growth rate being higher at higher temperature (Fig. 3). The intraspecific competition coefficients were negative for Pm I and Pm III at 15 °C. At 15 °C, Pm III had the lowest maximum growth rate, while Pm I at 20 °C had the highest maximum growth rate.

Figure 3 Estimated maximum growth rate (r) per species at two temperature regimes: 15 °C and 20 °C.

Error bars represent 95% confidence intervals.

The intraspecific competition coefficients (a) were negative for Pm I and Pm III at 15 °C (Fig. 4), suggesting that per capita growth rates increased with higher densities. This positive relationship between growth and density could indicate the presence of Allee effects for these species at 15 °C. The other treatments showed positive intraspecific competition coefficients varying between 1.54 × 10−4 ± 7.08 × 10−5 for Pm III at 20 °C and 0.07 ± 0.05 for Pm IV at 15 °C. While the effect of temperature alone was not significant, both species and the interaction between species and temperature had a significant effect on the intraspecific competition coefficients (two-way ANOVA: Temp: F = 1.44; p = 0.246, Sp: F = 8.82; p = 0.002, Temp × Sp: F = 12.61; p = 0.0003). For instance, Pm I exhibited greater coefficients at 20 °C (pairwise Tukey HSD: p = 0.013), contrasting Pm IV which showed higher intraspecific competition at 15 °C (pairwise Tukey HSD: p = 0.032). Pm III did not show a significant difference between 15 °C and 20 °C (Fig. 4).

Figure 4 Estimated intraspecific competition coefficient per species at two temperature regimes: 15 °C and 20 °C.

Error bars represent standard deviation of the parameter estimation.

The estimated parameters for Pm IV showed a notably high standard deviation for both maximum growth rate and intraspecific competition coefficients. Likewise, the intraspecific competition coefficient of Pm III at 15 °C.

Interaction experiment

Population abundance in the interaction treatments

The interaction (combined species) experiment was sampled and analysed after 8, 16 and 24 days. Data of some replicates and time points were discarded because qPCR yielded values above the cycle threshold, while some time points (C34 at 15 °C at 16 days and C14 at 20 °C at 24 days) had no valid results for any of the replicates.

Species performance varied per species combination and temperature (Fig. 5). All species combinations and temperature treatments, except for the combination of Pm I and Pm IV (C14) at 20 °C, clearly revealed a dominant species in each assemblage at the end of the experiment. For the assemblage of Pm I and Pm III (C13), temperature defined the dominant species: Pm I was the strongest competitor at 15 °C while Pm III at 20 °C. For the other combinations, Pm I and Pm III achieved higher abundances than Pm IV at all species combinations and temperatures. Despite a generally clear dominance of one species over the other, no competitive exclusion was observed, except in assemblages of Pm III with either Pm I or Pm IV at 20 °C, where Pm I and Pm IV were excluded, respectively.

Figure 5 Abundance of adult individuals per species over time in combined species treatments.

The number represents species combinations as C13 (top), C14 (middle) and C34 (bottom) at two temperature regimes: 15 °C (left) and 20 °C (right). Abundances are represented by recorded data (data points) and the fitted monospecific or competition model (lines). Error bars represent 95% confidence intervals of 4 replicates.

The Lotka–Volterra model (Eqs. (2a) and (2b)) showed a very good fit for all species combinations at 15 °C (NRMSE < 0.1), but less so at 20 °C, especially for the pair of Pm I and Pm IV (C14, NRMSE = 0.74; Fig. 5). The experimental data for the combination of Pm I and Pm IV fitted the Lotka–Volterra model the worst (NRMSE = 0.549).

Figure 6 shows the effects of temperature on the final adult abundance of both monospecific and combined-species cultures of each species. For Pm I and Pm III, temperature had a strong effect on both monospecific and combination treatments (two-way ANOVA: Temp effect: p < 0.05 for both species; p > 0.05 for PM IV; Table 2). For Pm I, all populations reached higher abundances after 24 days at 15 °C than at 20 °C. Conversely, the abundance of Pm III after 24 days was higher for all populations at 20 °C than at 15 °C (Fig. 6). While the monospecific populations of Pm IV on average grew better at 20 °C, both combined treatments reached higher (average) adult abundances after 24 days at 15 °C compared to 20 °C. Due to the high variability among replicates of Pm IV, these differences were not significant (p > 0.05; Table 2). Along with temperature, the particular species combinations can also affect population development. At 15 °C, for example, Pm III had a strong positive effect on the population development of Pm I (p = 1.41 × 10−5). At the same temperature, the presence of Pm IV had no significant effect on the population of Pm I (p > 0.05; Table 2).

Figure 6 Total abundance of adult individuals in monospecific (M1, M3, M4) and combined treatments (C13, C14, C34) at two temperature regimes (15 °C and 20 °C) for the species Pm I (top left), Pm III (top right) and Pm IV (bottom).

Error bars represent 95% confidence intervals.

Table 2 Results of two-way ANOVA on total abundance of adult nematodes per species for monospecific and combined treatments and Tukey HSD tests of the pairwise combination treatments.

Significant values are marked in bold.

	Pm I	Pm III	Pm IV	
	F	p-value	F	p-value	F	p-value	
Temperature	118.62	1.41E−07	111.54	1.98E−07	0.001	0.980	
Combination	29.49	2.33E−05	3.32	0.071	0.23	0.798	
Temperature × Combination	28.74	2.65E−05	8.66	0.005	2.61	0.114	
M-C13		0.0001		0.123		–	
M-C14		0.784		–		0.272	
M-C34		–		0.984		0.379	
C13-C14		4.18E−05		–		–	
C13-C34		–		0.093		–	
C14-C34		–		–		0.107	

The population development, estimated as average adult abundances over time, behaved differently for the different species and treatments at the two different temperature regimes (Fig. 7). While most populations showed signs of exponential or logistic growth, some showed an initial growth, followed by a decline and complete collapse after 24 days. Notably, both Pm I and Pm IV showed this decline when combined with Pm III at 20 °C.

Figure 7 Total adult abundance of all treatments per species: Pm I (top left), Pm III (top right) and Pm IV (bottom) at two temperature regimes: 15 °C and 20 °C.

Population growth model in the interaction treatments

Figure 8 displays the estimated interspecific interaction (competition) coefficients for the different species combinations and temperatures. Both competitive (positive coefficient) and facilitative (negative coefficient) interactions were present in the combination (interaction) experiments. A change in temperature often changed the interaction from facilitation to competition or vice versa.

Figure 8 Estimated interspecific competition coefficients per species interaction pairs at two temperature regimes: 15 °C and 20 °C.

Species interactions are indicated as number pairs in order of competitive success; e.g. a13 represents the competitive effect of Pm III on Pm I. A positive coefficient indicates competition, a negative coefficient indicates facilitation.

Since the competition model (Eqs. (2a) and (2b)) was fitted to averages of the population data, a large amount of experimental variability was lost. Therefore, these interspecific competition coefficients can only provide an indication of the interactions present in the experiments. Additionally, the interaction coefficients can only indicate the net positive or negative effects. In reality, both positive and negative effects are likely present to some extent. Thus, a positive interspecific competition coefficient means that the competitive effects outweigh the facilitative effects.

Different pairs of species showed different types of species interactions. Mutual facilitation was present in the combination between Pm I–Pm III at 15 °C. For other species and temperature treatments, one species facilitated while the second species was competing. Temperature appeared to influence species interactions differently. Pm I and Pm III showed facilitative effects at 15 °C, shifting towards competitive effects at 20 °C. By contrast, species paired with Pm IV experienced competitive effects at 15 °C and facilitative effects at 20 °C.

The average fitness ratio for the different species pairs is given in Fig. 9. If the ratio kj/ki is larger than one, species j is the strongest long-term competitor. In the Pm I–Pm III pair, Pm III was the superior competitor at both temperatures. Pm I performed better at 15 °C, making the competition more even (ratio closer to one). Similarly, the average fitness ratio indicated that Pm I was the superior competitor over Pm IV at both temperatures, but the kj/ki ratio was closer to one at 15 °C. For the Pm III–Pm IV pair, the superior competitor changed in function of the different temperature treatments: Pm IV at 15 °C, while Pm III at 20 °C.

Figure 9 Average fitness ratio (kj/ ki) of the three species pairs at two temperature regimes: 15 °C and 20 °C.

A kj/ ki ratio > 1 means species j is competitively superior.

Discussion

Our results showed that both competitive and facilitative pairwise interactions were present between three nematode cryptic species of the Litoditis marina species complex. Furthermore, temperature affected both monospecific population growth and the type and strength of the pairwise interactions between these species.

Effects of temperature on monospecific population development

The population development of all three tested species was strongly influenced by temperature. Generally, a higher temperature resulted in a higher maximum growth rate for all cryptic species.

Pm I reached higher maximal abundances at 15 °C, while Pm III and Pm IV at 20 °C. Previous research with the same cryptic species and similar conditions (media, timeframe), but without competition treatments, reported a similar effect of temperature on population development for Pm I and Pm III (De Meester et al., 2015a). The results of both studies also correspond to the seasonality in the field distribution of these species in the Westerschelde Estuary and along the Belgian coast: Pm III has a higher relative abundance (the abundance of one cryptic species over the total abundance of all cryptic species present) in summer, while Pm I shows a higher relative abundance in winter and spring (Derycke et al., 2006). For Pm IV, our results contradict the results from De Meester et al. (2015a), where Pm IV reached significantly higher abundances at 15 °C than at 20 °C.

Increasing temperature usually leads to faster development and shorter life cycles in nematodes, related to faster enzymatic reactions (Hopper, Fell & Cefaln, 1973). Furthermore, a higher temperature often leads to an increase in reproduction rates and reproductive output (De Meester et al., 2015a; Moens & Vincx, 2000a). However, this increased growth can only be sustained up until a species-specific threshold temperature, after which population growth will decrease and mortality will increase. It is possible that Pm I is nearing this threshold temperature in the 20 °C treatment since a strong population decline was apparent after only eight days for all replicates of the monospecific treatment and in alignment with personal observations. A different temperature response has been reported by two other studies (Moens & Vincx, 2000a; De Meester et al., 2015a), reporting that Pm I populations reached their highest abundance at 20 °C (Moens & Vincx, 2000a) or at 15 °C (De Meester et al., 2015a) and declining at 25 °C.

Another factor influencing these results could be local adaptation. Nematode populations with short generation times which are cultured for an extended period of time can impose a ‘natural selection’ for individuals performing better under the inoculation conditions. L. marina populations have been shown to adapt their environmental tolerances and life-history characteristics to specific culture conditions (Moens & Vincx, 2000b; Moens & Vincx, 1998; Vancoppenolle, Borgonie & Coomans, 1999). Therefore, the response of these cultured populations to temperature change may differ from the response of natural populations due to local adaptation to the culture’s optimal conditions. This can probably also explain the lower performance of Pm I at 20 °C in our experiments.

The estimated parameters for Pm IV showed a notably high standard deviation for both maximum growth rate and intraspecific competition coefficient. Likewise, the intraspecific competition coefficient of Pm III at 15 °C. Since these parameters were used as input for the estimation of interspecific competition coefficients, this larger uncertainty could affect the outcomes of this parameter estimation for species combinations with Pm III and Pm IV.

The negative intraspecific competition coefficients for Pm I and Pm III at 15 °C point to the presence of Allee effects. Allee effects occur when the fitness of a small or sparse population declines, as population size or density also declines. In other words, the Allee effects display a positive correlation between population fitness or growth and population density (Berec, Angulo & Courchamp, 2007). Multiple mechanisms can drive these Allee effects; one likely cause in these experiments could be the increased difficulty in finding a mate at a lower population density.

Effects of temperature on L. marina species interactions

The interaction experiment (combined species treatments) showed that temperature had a more pronounced effect on the population abundances of L. marina species than the presence of another species. Interspecific interactions, therefore, do not shift the dominance of species under stable conditions (here temperature). Different temperatures, however, affected differently the involved species and therefore, altered their species dominance. More specifically, Pm I reached higher abundances than the other species at 15 °C. At the higher temperature of 20 °C, the highest abundances were achieved by Pm III. Pm IV had the lowest abundances in every species pair for both temperatures (15 °C and 20 °C).

Previous multispecies experiments containing four L. marina cryptic species showed that at 20 °C Pm III was the best-performing species, while Pm IV was the worst-performing species and was even completely extirpated after 35 days (De Meester et al., 2011). A similar experiment with Pm I, Pm III and Pm IV at a constant temperature of 20 °C produced similar results (De Meester et al., 2015c). Pm III had the highest abundance, followed by Pm IV and Pm I. An experiment with only Pm I and Pm IV showed that Pm I achieved a higher abundance than Pm IV (De Meester et al., 2015c). This same trend is present in the current experiment, although a high intraspecific variability was observed.

The estimated competition coefficients indicated a mixture of facilitative and competitive pairwise effects, although only the facilitative effect of Pm III on Pm I translated into a significant effect on maximal adult abundances. According to the estimated competition coefficients, a temperature increase from 15 °C to 20 °C shifted the facilitative effects of Pm I and Pm III towards more competitive effects. On the contrary, the same temperature increase changed the competitive effect of Pm IV at 15 °C towards a more facilitative effect at 20 °C. Several mechanisms could lead to facilitation between nematode species. One mechanism for facilitation could be the effect of a larger initial nematode population on bacterial populations. Grazing by nematodes has been shown to increase or stimulate bacterial growth by maintaining the bacterial populations in the exponential growth phase. In this way, a higher initial nematode population could achieve higher food concentrations later in the experiment (De Meester et al., 2015c; Moens et al., 2013). Additionally, grazing as well as the production of mucus tracks while crawling within the agar, can shift the bacterial community composition (De Mesel et al., 2004; Moens et al., 2005; D’Hondt et al., 2018), thus increasing or decreasing its suitability for consumers.

Both competition and facilitation have been observed in earlier experiments on these species (De Meester et al., 2011; De Meester et al., 2015b; De Meester et al., 2015c). While the current experiments investigated only pairwise interactions between the cryptic species, previous experiments investigated multispecies cultures with three or more species in combination. For populations containing more than two species, there appear to be complex interactions, different from the sum of their pairwise interactions, described as a competitively intransitive network (De Meester et al., 2015c). Within this network, both the presence and the densities of one species can influence the interactions of the already present species (i.e., interaction modification) and directly or indirectly affect the community dynamics (De Meester et al., 2015c).

Species interactions like competition and facilitation play an instrumental role in the coexistence or exclusion of species from a community, and thereby in community structure and composition. Although our competition experiments lasted for a relatively short time (24 days), both Pm I and Pm IV were excluded after 16 days in the presence of Pm III in the cultures at 20 °C, thus confirming that Pm III is the strongest competitor at that temperature (Table 3). At 15 °C, on the other hand, two-species coexistence was possible for at least 24 days, albeit at low population abundances for one of the involved species (Table 3). Furthermore, the population dynamics in these experiments seem to not have reached (quasi-)equilibrium at both temperatures tested, indicating temporary coexistence at least until an equilibrium can be reached. This suggests the effects of temperature on species interactions may affect the possibility of coexistence or competitive exclusion between L. marina cryptic species. For Pm I, this population decline might not be related to competitive interactions, since it also displayed a strong population decline after 8 days and complete extinction after 24 days in the monospecific culture at 20 °C. Additionally, only data from the adult population was examined in this experiment, so it remains possible that some juveniles remained present but did not yet develop into adults. A strong temperature effect was observed for PM I, independent of competition, but its competitive abilities clearly depend on temperature.

Table 3 Summary of the outcome of species interactions of the combination experiments after 24 days.

Read per column for the main outcome per species in combination with the species in rows. Colors refer to different outcomes (i.e. bold, inferior competitor; underline, superior competitor; italic, extirpated; and bold and italic, equal competitor or facilitator).

Combination	Pm I	Pm III	Pm IV	
Pm I	15 °C	–	Inferior	Inferior	
20 °C	–	Superior	Equal	
Pm III	15 °C	Superior	–	Inferior	
20 °C	Extirpated	–	Extirpated	
Pm IV	15 °C	Superior	Superior	–	
20 °C	Equal	Superior	–	

The average fitness ratios suggest that with a 5 °C increase in temperature, Pm III would become a better long-term competitor than Pm I and Pm IV. In the absence of stabilizing forces, this could lead to the complete competitive exclusion of Pm I and Pm IV, which was present in the current experiments at 20 °C. While the combination of Pm I and Pm IV have not been found sympatrically in nature, this combination would likely result in Pm I being a better competitor at either temperature.

Species coexistence in a changing climate

A rise in average temperature and the higher frequency and intensity of marine heat waves due to climate change could strongly impact nematode assemblages and the dynamics between species populations, as shown in these experiments. Free-living nematode assemblages in nature are locally diverse and usually comprise several closely related (both phylogenetically and functionally) species.

The results presented here suggest that Pm III would benefit from a 5-degree temperature increase within the range of 15 °C–20 °C. Pm I, which is currently the dominant species in the Westerschelde estuary and along the Belgian coast, would likely suffer from such a temperature increase. Both temperature and competition impact the abundance of Pm IV, with a higher monospecific abundance at 20 °C, but higher abundance in the presence of other species (combination cultures) at 15 °C.

Along with these direct effects in nematode populations, temperature changes could influence nematode assemblages through indirect effects on food availability, and biogeochemical gradients, among others. Further, the effects of temperature variation on species interactions could change when three or more species co-occur (De Meester et al., 2015c), as species interactions become more complex. Temperature variation and the differences in temperature niche breath of the organisms play an important role in their coexistence dynamics, as demonstrated in studies on co-occurring soil nematodes (Anderson & Coleman, 1982; Sohlenius, 1985). Non-additive species interactions, which have been characterised as a non-hierarchical intransitive species network, as explained above, are often present within such a cryptic species complex (Rojas-Echenique & Allesina, 2011; De Meester et al., 2015c). The impact of a temperature increase could have differential effects on species fitness and assemblage, depending on the distinct combination of species, altering species dynamics and leading to the coexistence or exclusion of one or more species within this network.

Despite the close genetic relationship within the L. marina complex, notable functional distinctions among the species have emerged. Different species in this cryptic species complex show differences in selective feeding (Derycke et al., 2016; Guden, Derycke & Moens, 2021) and microhabitat preferences (Guden et al., 2018) and they have been found to differentially influence decomposition processes (De Meester et al., 2016). In another species complex, that of the marine nematode Halomonhystera disjuncta, cryptic species within the complex have been reported to show different responses/sensitivity to the presence of sulphides, in addition to temperature and salinity (Van Campenhout et al., 2014). Other significant functional roles of free-living nematodes, like their trophic roles, e.g., predation or top-down controls of microbes and protists, can also be impacted by changes in environmental conditions in a changing climate. The shifts in species interactions and community composition induced by climate change could therefore have cascading effects on ecosystem functions and services.

Furthermore, this experimental setting did not allow the possibility for dispersal, which may also change the species dynamics. Active dispersal has been found to be very species-specific (De Meester, Derycke & Moens, 2012; De Meester et al., 2015b; De Meester et al., 2018). At the same time, dispersal may also be triggered by inter- and intraspecific competition, thus the inferior species may have a greater dispersal ability (at a local scale), as a coping mechanism for the longer-term survival of the species at increased competition levels. Our knowledge of marine free-living nematode dispersal is generally limited (Thomas & Lana, 2011; Derycke, Backeljau & Moens, 2013; De Meester et al., 2018), however, it has been suggested that occupying a dispersal niche may allow competitively inferior species to persist alongside a superior species (Aiken & Navarrete, 2014). Other researchers suggested that priority effects play a more important role with temperature increase, which enhances interspecific differences in dispersal (Grainger, Rego & Gilbert, 2018).

Similarly, the diversity of microhabitats and the availability of substrates and food resources also affect the assemblages of these L. marina species and influence their coexistence (De Meester et al., 2015b; Guden et al., 2018; Guden, Derycke & Moens, 2021). In these experiments, a single strain of E. coli was provided as food for nematode populations, albeit in non-limiting quantities. More diverse food sources could differentially impact species fitness and community dynamics (Derycke et al., 2016; Guden, Derycke & Moens, 2021). Bacterial diversity and abundance are also highly influenced by temperature, thus an indirect effect on resource competition can be imposed by climate change. Moreover, fluctuating environmental conditions, along with predation and storage effects are also important factors that need to be taken into account for coexistence dynamics.

Conclusions

Changes to the environmental conditions can affect marine species and their communities. Temperature variation can also influence the interactions between species, leading to cascading effects on the abundance and composition of communities.

Our experiments indicate that temperature is not only a major factor influencing population growth and fitness of marine free-living nematodes but also affects the type and strength of the pairwise interactions between species. Our results show that both competitive and facilitative interactions are present between three species of the L. marina complex and that an increase in temperature can cause shifts in species interactions from competitive to facilitative, or vice versa, and could even lead to competitive exclusion.

The projected increase in sea temperature caused by global anthropogenic climate change will therefore likely alter nematode community abundance, composition and diversity. Such a temperature increase is expected to facilitate Pm III populations and inhibit Pm I populations in the Westerschelde estuary and along the Belgian coast. The species, despite being genetically very close, display different functions and spatiotemporal dynamics along the estuary. Consequently, such effects of temperature increase on community composition can have severe consequences for ecosystem functioning and ecosystem services.

Annelien Rigaux (Ghent University, CeMoFE) is acknowledged for her valuable contribution in conducting the experiments and the qPCR analyses.

Additional Information and Declarations

Competing Interests

Author Contributions

Data Availability

The authors declare there are no competing interests.

Anna-Maria Vafeiadou conceived and designed the experiments, performed the experiments, analyzed the data, prepared figures and/or tables, authored or reviewed drafts of the article, and approved the final draft.

Kevin Geldhof performed the experiments, analyzed the data, prepared figures and/or tables, authored or reviewed drafts of the article, and approved the final draft.

Wissam Barhdadi performed the experiments, analyzed the data, prepared figures and/or tables, authored or reviewed drafts of the article, and approved the final draft.

Jan M. Baetens conceived and designed the experiments, authored or reviewed drafts of the article, and approved the final draft.

Bernard De Baets conceived and designed the experiments, authored or reviewed drafts of the article, and approved the final draft.

Tom Moens conceived and designed the experiments, authored or reviewed drafts of the article, and approved the final draft.

Aisling J. Daly conceived and designed the experiments, analyzed the data, prepared figures and/or tables, authored or reviewed drafts of the article, and approved the final draft.

The following information was supplied regarding data availability:

The data is available in the Supplemental Files and at VLIZ Marine Data Archive: Vafeiadou, A-M.; Geldhof, K.; Barhdadi, W.; Rigaux, A.; Baetens, J.M.; De Baets, B.; Moens, T.; Daly, A.J.; Marine Biology Research Group, Department of Biology; Department of Data Analysis and Mathematical Modelling. Ghent University: Belgium; (2024): Experimental dataset of temperature-driven competition dynamics experiments. Marine Data Archive. https://doi.org/10.14284/656.

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
