# Peer review of "Temperature-driven dynamics: unraveling the impact of climate change on cryptic species interactions within the Litoditis marina complex"

_PeerJ, doi:10.7717/peerj.17324_

## Round 0.1 · original submission · Minor Revisions

Dear Dr. Vafeiadou,

Your manuscript has now been evaluated. As you will see the reviewers had positive comments about your manuscript but also provided important comments. Please consider the comments carefully and submit the final version.

Best regards,

·

Basic reporting

Well writting interesting paper. All the ideas are clearly explained and the results will be interesting not only to nematodes ecologists, but for wide renge of merine biologists.

Experimental design

Experiments are correct and the methods are properly described.

Validity of the findings

The interpretation of the results looks correct and being in accordance with data obtained. I have a minor comment here: authers stronlgy address their results to "increasing temperature caused by anthropogenic climate change". It repeats several times in the text. Hovewer, the results themselves show prominent effect of temperature variation regardless of the reasons for the changes. I suggest to be more precise in wording.

Additional comments

I do not see any obstacles to publish the manuscript "as it is".

Reviewer 2 ·

Basic reporting

no comment

Experimental design

no comment

Validity of the findings

no comment

Additional comments

Overall, the manuscript is very well written, and the language is clear and unambiguous. Figures, tables, and supplemental materials are relevant and well illustrate the research.
The structure of the manuscript and its design conform to PeerJ standards.
The methods are well described and do not raise any questions.
A few minor notes:
- Line 128 – the unit of measurement of salinity is not specified
- Reference Moens et al., 2000a not listed in references
- The text does not contain a reference to the work of De Meester N, Derycke S, Moens T, 2012. Differences in time until dispersal between cryptic species of a marine nematode species complex. PLoS One 7:1.8. Doi: 10.1371/journal.pone.0042674.

General remarks:
The experimental part of the work, in my opinion, was done very well. However, after reading the entire manuscript, one is left with the impression of a discrepancy between the title-objectives-results-discussion.
In particular, the title somewhat misrepresents the content of the work, focusing on the impact of climate change on species interactions. As a reader, I would prefer to see a title that gives a clear idea of the essence of the work, which, according to the authors, addresses two research questions: (i) Does temperature differentially affect population growth in co-occurring L. marina species? and (ii) Does temperature affect the interspecific interactions between L. marina species? The results of solving these questions can be used in predicting the dynamics of communities when temperature changes in different directions (increase/decrease), manifestations (seasonal, climatic, geographical...), and causes of temperature change. While the authors throughout the manuscript focus primarily on anthropogenically caused global warming, the presentation of the results appears incomplete and distorted.
Much of the discussion in the section "Climate change and future perspectives" is not directly related to the results presented. At the same time, the discussion contains virtually no comparisons with data from similar studies on representatives of other taxa, including nematodes. It would be interesting and appropriate to compare the results with publications made several decades ago, before the “era of anthropogenically caused global warming”, for instance:
- Anderson R.V., Coleman D.C. 1982. Nematode temperature responses: a niche dimension in populations of bacterial-feeding nematodes. Journal of Nematology. Vol. 14(1). P. 69-76.
- Sohlenius B. 1985. Influence of climatic conditions on nematode coexistence: a laboratory experiment with a coniferous forest soil. Oikos. Vol. 44. P. 430-438.
- …
Or with other taxa?:
- Grainger T.N., Rego A.I., Gilbert B. 2018. Temperature-dependent species interactions shape priority effects and the persistence of unequal competitors. The American Naturalist. Vol. 191(2).
- …
It seems to me that such comparisons would be interesting and appropriate. Despite the comments, the work leaves a very good impression and can be accepted with a Minor Revision.

Reviewer 3 ·

Basic reporting

This is a very interesting and well done article. It is convincing that, in general terms, (to summarise very much) the three species can be characterised as follows. Pm III: efficient competitor, loves temperature 20°C. Pm IV: weak competitor, loves 20°C. Pm I: intermediate competitor, loves 15°C.
Given that we are talking about intra- and interspecific competition (and facilitation), it would be interesting to know the average distance between one Litoditis individual and another, both in nature and in laboratory tanks.
I have not been able to find any errors, except for the following little things:
line 528: the cited author (Thomas 2011) does not appear in the References.
line 622: better write the surname Ysebaert
Table 3: I believe that the first line of the caption "Summary of the outcome... 24 days" is redundant, because a little further down it is repeated in the complete caption. This is not an error, but a useless line.
Figure 1: I think the first line of the caption "Maximal abundances... treatments" is redundant.
And so is the first line of captions for all the other figures.

Experimental design

Very good.

Validity of the findings

Surely valid.

---

## Round 0.2 · accepted · Accept

In the revised version the authors took into consideration all comments and remarks. I recommend accepting the manuscript for publication in PeerJ.